# Low-Code Machine Learning Platforms: A Fastlane to Digitalization

**Krishna Raj Raghavendran**  and **Ahmed Elragal** *

Department of Computer Science, Electrical and Space Engineering, Luleå University of Technology, 97187 Luleå, Sweden; krirag-1@student.ltu.se
* Correspondence: ahmed.elragal@ltu.se

**Abstract:** In the context of developing machine learning models, until and unless we have the required data engineering and machine learning development competencies as well as the time to train and test different machine learning models and tune their hyperparameters, it is worth trying out the automatic machine learning features provided by several cloud-based and cloud-agnostic platforms. This paper explores the possibility of generating automatic machine learning models with low-code experience. We developed criteria to compare different machine learning platforms for generating automatic machine learning models and presenting their results. Thereafter, lessons learned by developing automatic machine learning models from a sample dataset across four different machine learning platforms were elucidated. We also interviewed machine learning experts to conceptualize their domain-specific problems that automatic machine learning platforms can address. Results showed that automatic machine learning platforms can provide a fast track for organizations seeking the digitalization of their businesses. Automatic machine learning platforms help produce results, especially for time-constrained projects where resources are lacking. The contribution of this paper is in the form of a lab experiment in which we demonstrate how low-code platforms can provide a viable option to many business cases and, henceforth, provide a lane that is faster than the usual hiring and training of already scarce data scientists and to analytics projects that suffer from overruns.

**Keywords:** low-code; no-code; machine learning; auto ML; ML platform; data scientist scarcity; projects overruns



## 1. Introduction

In the contemporary data-driven world, any organization can have access to data, but how well they make use of it to generate insights will dictate their market position. To take a competitive edge over rivals, it is not just enough to have efficient transactional systems; it has become essential to analyze the historical data promptly and propose necessary actions for the business to take. Previous research has highlighted the significance of data-driven decisions and their potential to realize values over decisions based solely on opinions [1]. Other research has pointed out that understanding the data is key, and data science is the discipline that helps expand knowledge about available data and generate insights from these data. Furthermore, research explained that data science helps to attain data-driven decisions [2].

Data science is an umbrella term for various in-depth studies that can be classified into three major areas: data analytics, data mining, and machine learning (ML) [3]. The data science paradigm encompasses discrete roles and responsibilities, such as data engineers, data analysts, and data scientists, as well as external dependencies such as product owners, project sponsors, business analysts, IT managers, c-level executives, etc. A person with a specific persona does not need to be an expert in other trades or have a cross-functional skillset. However, it would be advantageous if a person with an understanding of business requirements and objectives had access to a platform that allowed them to perform the basic operations of data-science-related roles without prior coding, analytics, or data engineering experience.

On the other hand, Alsharef et al. emphasized that developing an ML model necessitates domain expertise and advanced ML programming skills [4]. They highlighted the difficulties in finding trained ML experts in the market; hence, automatic ML is seen as an asset that bridges the gap between data-science use cases and a lack of appropriate ML resources [4]. This is where the no-code/low-code ML platform comes into play. Before getting there, we first need to understand the evolution of ML and how we landed on advanced ML platforms with low or no code. Both low-code and no-code approaches aid in the rapid development of ML models, the automation of data pipelines, and the visualization of the findings. However, they differ greatly in terms of the type of audience willing to use this service. Developers can leverage existing building blocks and libraries while still having the flexibility to customize the task as required with the low-code approach. Conversely, no-code is primarily intended for domain experts with minimal to no prior software development knowledge [5]. With the no-code approach, users can use drag-and-drop functionality to execute the desired task, with minimal to no flexibility to customize. We can categorize cloud-native and cloud-agnostic ML platforms as low-code platforms since they allow us to build custom ML models by writing code in ML platform-native notebooks.

When it comes to developing ML models using the AutoML service, specifically, we must categorize it as a no-code since we anticipate the ML platform to conduct all the tasks in the ML lifecycle automatically with very few inputs from its users initially. Low-code ML platforms can be used by different personas, including data scientists and ML developers. Added to this, no-code AutoML services can also be used by persons with strong business or data domain knowledge, such as data engineers, data analysts, business analysts, or product owners. We can even form a cross-functional team comprised of all the aforementioned personas to create ML models using AutoML services; this, in the long run, would yield multiple benefits in terms of saving time and money. Research has supported the importance of emerging low-code cloud data platforms and their vital role in the speed of digitalization [6].

In this research, we examine similarities, differences, advantages, and limitations in leveraging some of the cloud-based low/no-code ML platforms. The Gartner Magic Quadrant published in 2020 for cloud-native AI developer service providers listed Amazon Web Services (AWS), Microsoft Azure, and Google Cloud Platform (GCP) as the top three leaders [7]. Hence, we chose these cloud-based platforms and their cloud-native ML services for further research. When it came to cloud-agnostic ML platforms, we chose Databricks for future investigation, as it is an enterprise-scale open-source unified data engineering, ML, and AI platform that is already integrated with all three cloud-native ML platforms listed [8]. These ML platforms can handle the entire ML lifecycle. In the following sections, we highlighted our findings by developing ML models without writing a single line of code using ML services available on the above-mentioned cloud platforms.

## 2. Problematization

Previous research has demonstrated the dire need to have a methodology for automatically selecting the optimum ML model and tuning the hyperparameters to improve ML model performance [9]. Building an ML model is quite a laborious task. ML developers must try out multiple algorithms and tweak hyperparameters constantly to derive the best ML model for solving a given business problem. This requires not only a thorough understanding of developing ML models but also time-consuming and intensive computing for data processing. Luo also emphasized the importance of the skillsets required for building state-of-the-art ML models manually, i.e., by a human attendant. Even with higher competencies, we cannot reduce the time spent refining the model and its hyperparameters to derive the best results [9]. While trying out different experiments to find the right model requires computation and substantial time, we may end up ramping up the computational resources as needed. The table below lists some of the ML algorithms and their corresponding hyperparameters.

From Table 1, we can understand that there are multiple hyperparameters connected to each ML algorithm. For decision trees, the max_depth parameter defines how far the leaf nodes can be split; when the maximum value set for this parameter is reached, it will stop splitting the node any further; the min_impurity_split parameter defines the minimum impurity level that can be at max up to the value set for this parameter; the min_samples_leaf parameter defines the minimum number of samples for each leaf node to be formed; and the max_leaf_node parameter defines the maximum number of leaf nodes we can have [9].

**Table 1.** ML Algorithms and their Hyperparameters.

| Algorithm | Hyperparameters |
|---|---|
| Decision Tree | max_depth, min_impurity_split, min_samples_leaf, max_leaf_nodes |
| Random Forest | n_estimators, max_features |
| Support Vector Machine | kernels, penalty value [3], tol |
| K-Nearest Neighbor | n_neighbors, metric, weights |
| Naïve Bayes | kernel density estimator, window width |
| Stochastic Gradient Boosting | learning_rate, n_estimators, subsample, max_depth |
| Neural Network | no. of hidden layers, no. of nodes in hidden layers, activation function, no. of epochs, learning rate |

For random forest, n_estimators define the number of decision trees to be generated, and max_features define the maximum number of features to be selected for each split.

For the support vector machine, the kernel defines how the input data will be represented, the penalty value is a regularization constant, and the tol parameter defines the stopping criteria for the model when no significant improvements are noticed on two consecutive iterations of training the model [3].

For k-nearest neighbor, n_neighbors define how many neighbors should be related, and the metric parameter defines the distance metric, for example, Euclidean distance.

For Naïve Bayes, the kernel density estimator defines the kind of data distribution to be considered, and window width is used for smoothing the kernel window size.

For stochastic gradient boosting, learning_rate defines how fast the ML model should learn and understand the pattern of the given data distribution; n_estimators define the number of trees or steps; subsample defines the subset of data to be considered; and max_depth defines the maximum depth of each tree [9].

For a neural network, we must find the ideal number of hidden layers, how many nodes should be present in each hidden layer, what would be the activation function, the number of epochs for trying out the maximum number of training iterations, and finally the learning rate [9].

Previous research on tuning hyperparameters for deep learning models by implying different optimization techniques symbolizes the complexity and expertise needed in transferring the previous learning to every new iteration of testing ML model performance [10,11]. Another research highlighted that tuning hyperparameters is time-consuming [12]. They also support the notion that finding the optimal hyperparameter value for an ML model requires multiple iterations of testing. Although not covering all parameters, Table 1 shows how complex it would be to select the best algorithm to address a business case by trying out different ML and ensemble models. This requires in-depth knowledge to address questions such as: Which type of ML algorithm to use? How to configure the hyperparameters? How to evaluate the model? How to select the best model? How to deploy the model to a different endpoint? Such a list is not comprehensive; however, the list can go on and on based on the type of business case we are trying to achieve. Another important aspect is how fast these questions can be answered, because

time is an important factor when considering market competitiveness. Additionally, to train and test multiple models, there is a need for scaling the computational resources. This is where cloud-based ML platforms come into the picture, which can address all the questions easily and in less time. Another advantage of a cloud-based ML platform is that we are not required to be masters of all trades; we could just have basic knowledge about data and business use cases and still be able to develop a classic ML model using the automatic ML features offered by different cloud vendors. Lastly, when the model is being trained, resources are scaled automatically in real time based on the requirements. As highlighted by Bahri et al., the automatic ML service helps in choosing the best ML model and tuning hyperparameters through multiple iterations of testing and different combinations of values [12].

## 3. The Methodology

Obtaining access to historical data to generate insights is an important question, especially for organizations pursuing their digitalization journey. Among the challenges they face is resource scarcity. To address such a challenge, different cloud platforms started offering managed services in order to handle sheer amounts of heterogeneous data and auto-scale the underlying infrastructure resources from which to process, analyze, and generate insights. Accordingly, we now have cloud platforms to handle big data, which provide capabilities to perform advanced ML tasks and can be used by different personas from non-programming backgrounds. In this research, we investigate the plausibility of that assumption, which entails the possibility of ingesting the data, developing and training the ML model, choosing a relevant model, evaluating the performance of the model, and eventually deploying it to the desired endpoint without writing a single code. Hence, we posit the following research question: "*How can low-code machine learning platforms provide a fastlane to digitalization?*"

In order to answer the research question, we adopted a mixed approach for conducting the research, where we used qualitative interviews with industry experts in order to conceptualize the problem with evidence from the field as well as a lab experiment in order to demonstrate the value of low/no-code platforms. Considering that our objective is to describe the current capabilities of different ML platforms and compare their similarities and differences, we therefore acquired the necessary data and applied a single ML scenario to them. Our lab experiment supports the feedback that we received from five expert interviews. Those we interviewed have used different cloud-based ML platforms and have come across real-world use cases from diverse industries.

## 4. Conceptualizing the Problem via Qualitative Interviews

We also conducted qualitative interviews with interviewees from five different SMEs working on ML and data engineering. The SMEs belong to different industries, e.g., AI consulting, reflecting their versatile backgrounds. Table 2 summarizes the interviewees' metadata.

**Table 2.** Interviewees' Metadata.

| Role | Years of Experience |
|---|---|
| Data Scientist | 10+ |
| Data Engineer | 7+ |
| Data Engineer | 10+ |
| ML Developer | 7+ |
| Data Scientist | 6+ |

We conducted qualitative interviews by asking semi-structured open-ended questions to the interviewees. All the interviews were conducted from a distance through Microsoft Teams. We conducted the interviews individually with each interviewee. Questions were not shared prior to the interviews with the interviewees, but the context of the interview was shared. The rationale behind choosing data engineers along with data scientists and ML developers is attributable to the responsibility that they have for setting up the whole data platform for ingesting and transforming the data so that they can be consumed by the data scientists and ML developers. Hence, we thought it was worthwhile to take the data engineer's input on this subject as well. The data scientist and ML developers are chosen for the most obvious reason, that they are responsible for choosing relevant ML algorithms, building ML models, evaluating the model results, choosing the best model, and eventually deploying it in the production environment.

We have asked those whom we have interviewed about the elements that ML platforms ought to have or provide, as well as the rationale for supporting such platforms. We summarize their responses to our questions as key pointers and segregated them into three major themes: AutoML-Centric, Human-Centric, and ML Platform-Centric, see below:

AutoML-Centric:

- The following tasks are recommended to be automated: data ingestion, orchestrating the ML model, and enabling auto-scaling to generate a production-ready ML model. Apart from this, any help for auto-tuning hyperparameters and choosing the best ML models is highly appreciated by the data scientists.
- Generating an automatic ML model can generally be used for faster go-to-market needs or proof of concepts.
- AutoML should work better on a smaller dataset. It will also be very useful for basic ML problems such as regression, classification, and time-series prediction.
- We should not completely rely on the results obtained from ML platforms and then act. We should set acceptance criteria to validate the results based on relevant metrics.
- We must have a complete grip on the values we are providing as input parameters, as they tremendously affect the ML model's performance. A small error in input could easily lead to highly biased results.

Human-Centric:

- ML platform services would help in bringing standardization to the way ML models are generated; otherwise, different ML developers may generate ML models in their own style, which may eventually cause trouble in maintaining respective ML models when they are deployed in production.
- Explaining the ML model is one of the most difficult tasks for the ML developers, whereas ML platform services come up with an auto-explainability feature off-the-shelf for any ML model generated.

ML Platform-Centric:

- Cloud-agnostic ML and data-engineering platforms can provide higher performance at a low cost.
- Cloud vendors must provide ML-optimized hardware for building an automatic ML model. They ought to provide an option to choose between standard and ML-optimized compute for processing the data.
- ML platforms must allow users to try out different ML models, validate the results, and choose the best ML automatically.
- ML platforms should take care of feature selection and feature extraction-related tasks, which demand significant human time and competence.

- ML platforms should allow users to try the most complicated deep learning algorithms with minimal code or fewer inputs. It should be possible to create a multi-layer neural network and tune it with less human interference. However, it should be created in accordance with the pre-condition criteria set by the respective ML platform vendors; thus, it is prone to improve over time. Hence, it is better to try this option than not try it out due to not having the required knowledge.
- Operationalizing the ML platform models should be made possible with the MLOps service available across the platforms.
- Continuously monitor ML model performance by comparing the results with the baseline model through the MLOps service, and it is also possible to trigger corrective action in terms of retraining the ML model when there is any data drift (when significant variation in the live data is found in comparison to test and validation data).
- There is always a cost involved while using ML platform services; hence, we must be careful and aware of which service we are leveraging and for what purpose.

## 5. Cloud-Based ML Platforms

Regarding cloud-based ML platforms, the three cloud vendors (AWS, GCP, and MS Azure) provide different services to address the use case of building an end-to-end ML model lifecycle without writing a single line of code and providing the least number of inputs. This helps business stakeholders who do not possess prior programming knowledge to be able to develop ML models easily.

In Figure 1, we have depicted the ML architecture based on the Azure ecosystem. Azure Data Lake Generation 2 can act as a data warehouse for storing structured, semi-structured, and unstructured data. We can even store all types of data in Azure blob storage. However, if we intend to use Azure Synapse, then it is a prerequisite to use Azure Data Lake instead. Regarding data transformation requirements, Azure offers the Synapse service, which acts as a lakehouse. This means it has the capabilities to store the data in conjunction with Azure Data Lake and is yet able to query using transact-standard query language (T-SQL) upon the metadata of data stored. Azure Synapse also supports atomic, complete, isolated, and durable (ACID) transactions. Synapse also consists of different features, such as data from different source connectors that can be ingested into Azure using linked services. Similarly, we can transform the data using some of the operations connectors in the pipeline. Synapse can invoke the Azure ML service for building manual or automatic ML models. Azure offers an ML service for building the ML model either using a pre-built model, through notebooks, or via the AutoML option. When the best model is built and evaluated, it is ready to be deployed at an endpoint. This is where the Azure container registry service comes into play; it takes care of containerizing the ML model and saving the container image, which can then be deployed using an orchestration service built by Azure, which is the Azure Kubernetes service. When the model is built and deployed on the endpoint, it will be continuously observed by Azure Monitor. When the model performance has decreased, due to significant change in the underlying trained dataset, this would trigger auto-training of the ML model again. If the auto-trained model has a low performance score, it is time to build a new or ensemble model. All the users registered with Azure Active Directory when they login to the Azure portal once will not be prompted again to access any other service to which they have access until they log out of the portal or time out due to being idle for a long time. Further, all the sensitive assets, such as passwords, authentication, or access keys, can be stored in the Azure key vault. Only the users having access to the key vault can assess the secrets stored inside it [13–15].

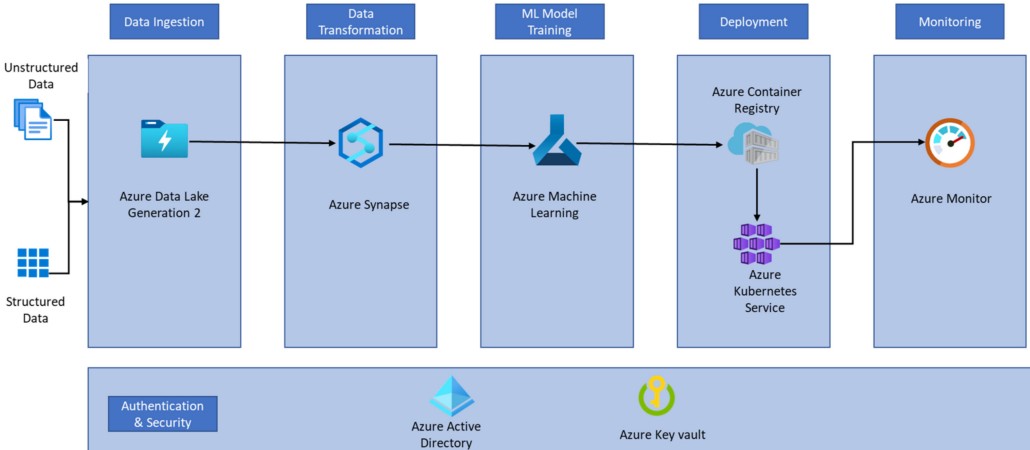

**Figure 1.** Azure native services offering holistic ML platform experience for the users (Adapted from [13]).

In Figure 2, we have depicted the ML platform architecture in the AWS ecosystem. AWS supports different types of data from heterogeneous sources. Data should first be uploaded into the Amazon S3 bucket or Amazon EC2 instance. When the data are within the AWS premise, they can be transformed using Amazon SageMaker Studio. AWS has developed SageMaker as a unified ML platform that can handle the end-to-end ML lifecycle. If the requirement is to generate automated ML, then we can make use of SageMaker's AutoPilot service. The AutoPilot takes care of training the model, evaluating the model, and choosing the best model based on the evaluation metrics score. When the best model is identified and tested, we can register it in the Amazon elastic container registry, then we will have a container image of our model that can be deployed to any endpoint. The Amazon Cloud Watch service is used to monitor AWS services. The Amazon single sign-on service is used to authenticate users to the AWS portal; once a user signs into the portal, they will not be prompted again to login when they access any of the AWS services to which they have access. The Amazon IAM service takes care of granting required privileges on resources to a role or a user [14,16,17]. Researchers [18] have explained the two phases of an AutoPilot job as candidate generation and candidate exploration. The candidate generation phase is responsible for splitting the dataset into train, test, and validation, exploring the data distribution, and performing necessary pre-processing. The candidate exploration phase is responsible for tuning different hyperparameters and finding the right values based on model performance metrics.

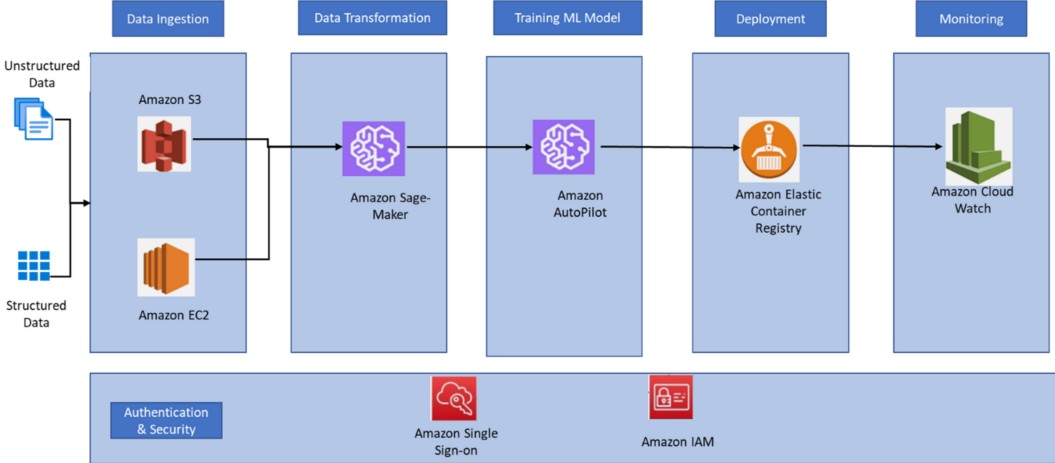

**Figure 2.** AWS native services offering holistic ML platform experience for the users (adapted from [14–16]).

In Figure 3, we have depicted the ML platform architecture based on the GCP ecosystem. Like Azure and AWS, GCP supports all types of data. The prerequisite for generating an ML model is that we upload the data to Google Cloud Storage and create a dataset. Google has developed vertex AI for the unified ML platform experience. We can perform all the data transformation tasks from vertex AI by creating data pipelines with the help of native data-engineering task templates. For generating ML models automatically, we can make use of the Google AutoML service. The AutoML service trains, evaluates, and chooses the best model automatically without writing a single line of code. When the model is ready, we can register it with the Google container registry. Then, the container image can be deployed to an endpoint using the orchestration service called Google Kubernetes. The Google monitoring service monitors all Google resources and triggers auto-healing when required. For identity and access management, we can use the cloud IAM service. For storing the secrets, we can use Hashicorp Vault integrated with GCP [14,16,17,19]. With AutoML in GCP, when it comes to image data, it could belong to any of the following categories: single-label classification, multi-label classification, object detection, and segmentation. With tabular data, we can choose between either regression, classification, or prediction. For natural language processing (NLP) business cases and text data, we can choose between the following categories: single-label classification, multi-label classification, entity extraction, and sentiment analysis. For video-related data, we can choose between the following categories: action recognition, classification, and object tracking.

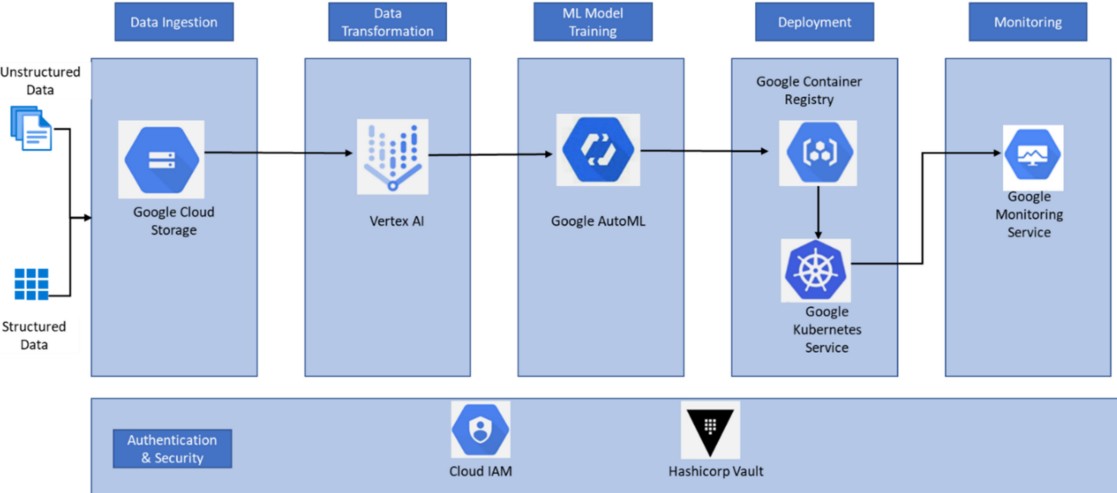

**Figure 3.** GCP native services offering holistic ML platform experience for the users (adapted from [19]).

All the cloud-based ML platforms provide respective cloud-native security, monitoring, and deployment solutions. In principle, they all support identity and access management for granting role-based access and have vault services for storing the secrets, access keys, and certificates.

In Table 3, we present a summary to compare the three platforms (AWS, GCP, and MS Azure).

**Table 3.** AI and ML Services offered by Azure, AWS, and GCP (adapted from [15]).

| AI & ML Use-Case | AWS | GCP | Azure |
|---|---|---|---|
| ML Platform | Sagemaker | Vertex AI | Synapse |
| Computer Vision | Amazon Rekognition and Lookout for Vision | Vision AI | Azure Cognitive Service Computer Vision |
| AutoML | Sagemaker AutoPilot | Vertex AI AutoML | Azure Machine Learning Service—Automated ML |

**Table 3.** *Cont.*

| AI & ML Use-Case | AWS | GCP | Azure |
|---|---|---|---|
| ML Frameworks Supported | TensorFlow, PyTorch, Apache MXNet | TensorFlow, PyTorch, Scikit-Learn | TensorFlow, PyTorch, ML.Net |
| NLP Service | Amazon Comprehend | Natural Language AI | Azure Cognitive Service Text Analytics |
| Speech to Text | Amazon Transcribe | Speech-to-Text | Azure Cognitive Service Speech to Text |
| Text to Speech | Amazon Polly | Text-to-Speech | Azure Cognitive Service Text to Speech |
| Language Translation | Amazon Translate | Cloud Translation | Azure Cognitive Service Translator |
| Conversational Service | Amazon Lex | Dialogflow | Azure Bot Service |
| Text Extraction | Amazon Textract | Document AI | Azure Form Recognizer |
| Recommendation & Personalization Service | Amazon Personalize | Recommendations AI | Azure Cognitive Service Personalizer |

## 6. Cloud-Agnostic ML Platform

Databricks is a cloud-agnostic data-engineering and ML platform. It follows the data lakehouse architecture. Traditionally, we have had enterprise data warehouses and data marts to store structured data, through which we can support business intelligence and reporting use cases. Later, with the help of different data lakes across all the major cloud ecosystems, we can ingest semi-structured and unstructured data as well. By doing so, we can support ML and data-science-related use cases. It is also common that both enterprise data warehouses and data lakes support various organizational use cases, forming a hybrid data architecture. The problem with hybrid architecture is its high complexity for administration and maintenance. Additionally, for different use cases, we must log on to different systems. To solve these problems, Databricks has introduced data lakehouse architecture, where we can still have data lakes from any cloud ecosystem of interest, and on top of the data lakes, we have delta lakes. The delta lake acts as a query engine, and it supports ACID transactions. We can use native SQL or Spark commands to query against the dataset available in the data lake more consistently and efficiently [20]. See Figure 4 for an explanation.

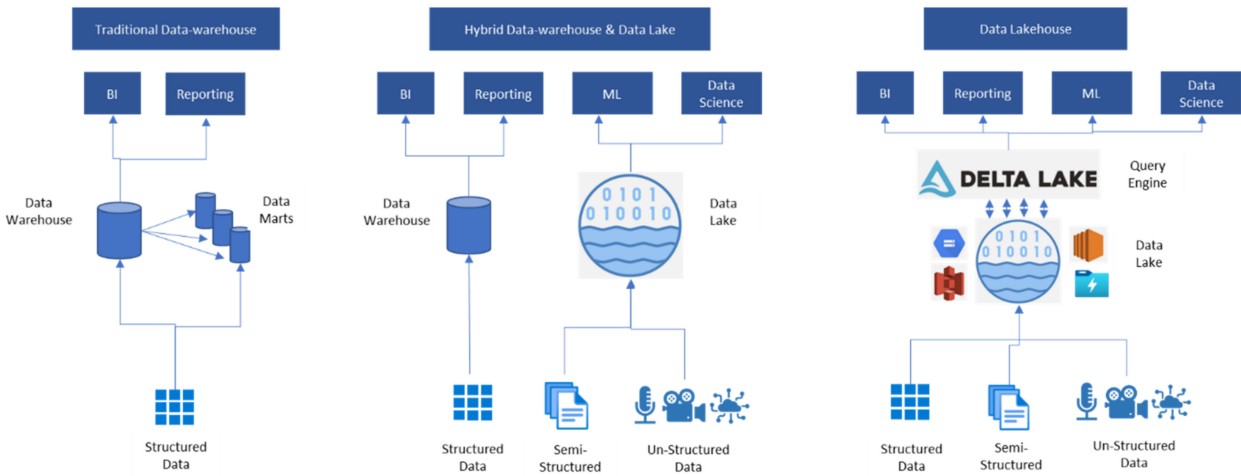

**Figure 4.** Cloud-agnostic Databricks Lakehouse Architecture (adapted from [20]).

Databricks can ingest data from any of the popular cloud ecosystems or streaming services. Data can be persistently stored in the cloud storage of choice. When the data are uploaded to cloud storage, we can create a dataset and consume them for building automatic ML models. Data processing and transformation are performed by the optimized Databricks Apache Spark runtime, which is specially designed for this purpose. To build an AutoML model, we must create Databricks clusters with ML-supported Databricks units (DBU) for computing. When the data and cluster services are available, we can execute the AutoML task to generate the ML model. We can monitor the model performance from experiments; similarly, clusters and other data engine assets/tasks can be monitored through Databricks command-line interface (CLI) or graphical user interface (GUI). We can register a git repository with Databricks repos to take in source code from the repository and deploy it in Databricks. When we create a cluster, we have the possibility of configuring the automatic scaling option, how many minimum worker nodes should be assigned to the task, and the maximum number of worker nodes. Similarly, we can set a timeout period, and the cluster will automatically shut down when it reaches the maximum timeout period set by us. When we create a Databricks instance, we must choose the pricing tier. If we choose a standard tier, then our Databricks environment would be secured by Apache Spark along with a cloud-specific active directory. If we choose the premium tier, then we obtain role-based access controls to provide fine-grained access. Data in the persistent storage layer are secured and protected through security services offered by the respective cloud vendors [8,21]. Researchers [22] pointed out that Apache Spark is the best-in-class in-memory data distributed framework. Databricks has optimized a version of Apache Spark that helps it process large datasets. In the Figure 5, we can see the different native databricks services available for building and monitoring the ML model.

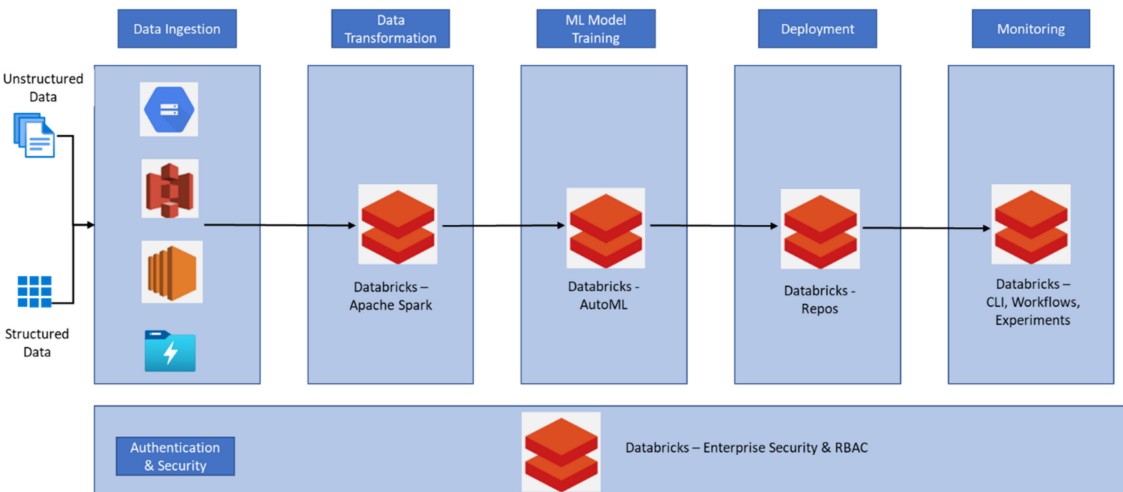

**Figure 5.** Databricks native services offering holistic ML platform experience for the users (adapted from [8]).

Previous research [23] reviewed building an advanced ML model manually and using AutoML vision to detect breast cancer with ultrasound scan results. They found that the AutoML vision service from GCP selected relevant ML models, such as random forest and convolutional neural networks, for this classification problem. They highlighted the ease of use while leveraging the AutoML service as a vital advantage. Other research [4] has underpinned the importance of AutoML for solving time-series-related forecasting. Time-series forecasting requires in-depth competence over simple linear to complex neural network models, especially tuning hyperparameters, which involves state-of-the-art skills. They [4] highlighted that this is where AutoML comes into play and can make a notable positive impact in auto-selecting the best ML models and auto-tuning the hyper-parameters.

Other use cases have started taking advantage of the AutoML feature, to name just a few, e.g., fraud detection, sales management, and customer experience [24].

## 7. Experimental Results

We studied three cloud ML platforms in this research, and all our further findings are connected to these cloud platforms. In the following section, we present our findings from each cloud platform and will descriptively compare the results. One thing to keep in mind here is that our research is focused mainly on generating automatic ML (AutoML) models without writing a single line of code or with low-level code. Even though it is possible to use other data formats, we chose tabular data for further analysis for simplicity and practical reasons.

### 7.1. Dataset Description

We have downloaded the Melbourne housing dataset from Kaggle (Link: https://www.kaggle.com/datasets/dansbecker/melbourne-housing-snapshot?select=melb_data.csv, (accessed on 22 April 2022)). The dataset consists of 21 features, of which 20 are descriptive and 1 is a target feature. The target feature is the property price. This dataset is not balanced and consists of missing values. We have not treated this dataset by performing manual pre-processing tasks or balancing the data distribution. We have ingested the raw dataset as-is into all three ML platforms and derived results. All the pre-processing tasks are automatically handled by the respective ML platforms.

### 7.2. Low-Code ML Platform Based on AWS

Amazon has published an open-source AutoML library called AutoGluon. Through which developers can create ML models for tabular, text, and image data with just a few lines of code. In addition, Amazon has developed an in-house fully managed ML service called Amazon Sagemaker.

In our case, we used the Sagemaker autopilot service to create the automatic ML model. The prerequisite for using the autopilot service is to upload the data to an Amazon S3 bucket so the Sagemaker autopilot service can consume the data for further analysis. Creating a new ML project is called an experiment in Sagemaker's terminology. We first uploaded our dataset to the Amazon S3 bucket. Then, we created a new experiment, where we provided the experiment name, chose the S3 bucket name, and picked our dataset from the list. Then, we provided the target feature name; in our case, we chose the feature price. Then, it is possible to deploy the best model automatically to the desired endpoint by enabling the auto-deployment feature and specifying the endpoint name or leaving the default name. It is also possible to provide the output directory name, which must be present in the S3 bucket where all the autopilot output logs will be stored. The above-mentioned options are the basic setting options; they are enough to create an auto-ML model. However, we have the possibility of restraining the ML generation behavior by tuning/tweaking the advanced setting option. We can define the following vital options as part of advanced settings:

- ML problem type: we can choose the following options: auto, binary classification, multi-class classification, and regression.
- Experiment run type: we can choose between executing the whole experiment or copying the generated code into a notebook and executing the commands cell-wise.
- Runtime: we can define how long the experiment can execute, how many maximum models it can generate, and the maximum time it can spend generating each model.
- Access: we can restrict access to any IAM role.
- Encryption: we can enable encryption for data present at the S3 bucket level.
- Security: we can use a virtual private cloud connection if we desire to have a highly secure private connection.

When we initiate a new experiment through autopilot, it automatically takes care of the following tasks: pre-processing, candidate definition generation, feature engineering, model training, explainability report generation, insights report generation, and the option to deploy the model to the desired endpoint.

The AutoPilot job generated different ML models and chose the best model with the least mean squared error (MSE). This model is built on the XGBoost algorithm. It took about two hours to generate all the models and choose the best model from the pool. The best model was automatically deployed to the endpoint specified. When we navigated to our model, it gave us richer information to understand the output results. It provided details on the explainability of the model, performance metrics, artifacts, and endpoints. The AutoPilot job also generates the feature importance based on the best model.

We noticed that distance, type of property, and number of rooms are considered the most important features of this model. As part of the automatic model build, AutoPilot automatically tested and tweaked hyperparameters to generate the best model.

There exists a list of artifacts generated from the AutoPilot job, which includes the input dataset, split of the training and validation sets, preprocessed training and validation sets, Python code for the feature engineering task, zipped folders consisting of all the feature engineering models, ML algorithm models, and other explainability artifacts. All the output data are stored inside the directory name that we specified earlier during experiment creation.

### 7.3. Low-Code ML Platform Based on GCP

We used GCP's Vertex AI for generating the AutoML model. It is a prerequisite that we have our dataset within Google Cloud for models to consume. Hence, the first step is to upload the dataset from the local machine to the Google Cloud. It is a mandate to create a dataset in vertex AI if we want to create a new model for the dataset. While creating the dataset, vertex AI can fetch the data from the local machine, Google Cloud, or a big query. However, it will create a dedicated directory within Google Cloud to store the dataset.

After creating the dataset, we can start training the model. While creating a training model, we must first choose the dataset that has been uploaded to GCP Storage. Based on the type of dataset, we will be given the option to choose the objective of the business problem. In our case, as we have tabular data, we are presented with regression or classification options, and we chose regression. Then, we also have the option to choose whether the model should be created automatically without any interference from humans or whether a pre-built model based on TensorFlow, Scikit-Learn, or XGBoost frameworks should be used.

In the next step, we must provide a name for the new model, and we also have the option to either create a new model or retrain an existing model. Then, we should choose the target field; in our case, we chose the price feature. When it comes to splitting the data, Vertex AI provides us with three different ways to split the data. The first option is to choose the data for training, testing, and validation at random; the second option is to choose them manually; and the third option is to choose the data in chronological order: the first 80% would be assigned to training; the next 10% would be assigned to validation; and the last 10% would be assigned to the test set.

In the next step, we can define different training options, such as changing the data type of a feature that is auto-detected or excluding a feature from further analysis.

We also have the option of adjusting the weight of the dataset for all the features based on the weight of a particular feature in the dataset; if not, by default, equal weight will be assigned by AutoML to balance the dataset. Then, we can optimize the training model based on RMSE, MSE, or RMSLE. RMSE can be chosen if we intend to give high importance to extreme values; MSE can be chosen if we intend to exclude extreme values as outliers; or RMSLE can be chosen if we intend to penalize error based on the relative weight.

As the last step, we have the option to choose the maximum node hours for training the model. The minimum number of hours that can be chosen is one, and we can choose a

higher value based on our requirements. Based on the value, the model will be allowed to train by autoscaling the required computing resources. With this, we can train a new model or retrain an existing one. With the four steps mentioned above, we can create a new model and train it without writing a single line of code. Model training will be allowed to execute until the budget node time is specified, and then it will automatically be stopped; no intervention is required.

When the AutoML job has generated new ML models, we can see additional details such as when the model training has started and until when it is allowed to execute; on which region compute resources were allocated for training the model, type of encryption key, dataset details, and data split details; whether we have trained the model with custom-built or AutoML; and, finally, what type of problem we are trying to address; in our case, a regression problem.

The trained model also generated the feature importance matrix. As per feature importance, we noticed that region name, land size, distance, and type of property are considered the most important features in deciding the price of the property.

We have the option of exporting our model as a TensorFlow-saved model docker container. By creating the model as a container, we can deploy it elsewhere promptly. We can also directly deploy our model at any desired endpoint as we wish. When the model is deployed to an endpoint, we are given the option to test our model from the respective endpoint without any need for manual testing, creating test strategies, or creating test cases. We also have the option of performing predictions in batches and storing the results in the specified cloud storage directory.

*7.4. Low-Code ML Platform Based on Microsoft Azure*

Microsoft Azure offers a unified ML platform experience through two of their major services: Azure Synapse and Azure Databricks. Azure has incorporated an optimized version of the Databricks Spark engine and called it Azure Databricks. Azure Synapse is an end-to-end ML platform, through which we can perform entire ML life-cycle tasks. Azure Synapse depends on the Azure ML service when it comes to creating ML models, training them, and evaluating their performance. The Azure ML service offers AutoML features, through which we can create a new model or retrain an existing one without writing a single line of code. Azure Synapse requires the Azure data lake to store the data; it does not just serve as a data warehouse but also offers to query against the underlying data. However, the Azure ML service requires either a compute cluster or a compute instance to process the data and train the model. Later, it also requires a compute cluster or compute instance for explaining the model, as this task also requires computing power. If we have a heavy-lifting task or a large dataset to analyze, then we can consider a compute cluster, which is a collection of interconnected nodes or instances. If we have a smaller dataset or a less resource-intensive task, we can consider a compute instance, a single node instance.

Before creating the automated ML model, we must create a persistent dataset in the Azure workspace blob storage by uploading the dataset from the local machine. Once the dataset is available in Azure blob storage, we can create an AutoML job. The first step is to choose the source dataset. Then, provide a name for the experiment and choose the target feature name; in our case, we chose the feature price. We must also choose the compute type for creating and training the new model. We can choose between a compute cluster or compute instance as per the business requirement; if none exists, then we must create one by choosing different available sizes of pre-built compute instances and clusters. The next step is to choose the type of task, whether regression, classification, or time-series forecasting. In our experimental case, it is a regression model. The last step is to select the validation type, where we can choose between auto or manual options. Then, we have the option to choose how to split the data for the test set. We have three options to choose from: either we can provide our own test dataset, skip the test dataset, or provide the percentage of data that should be allocated to the test dataset. In our case, we chose 20% of the data to

be allocated to the test dataset. The AutoML job generated multiple models and chose the best model based on the RMSE score.

The AutoML job also generated the feature importance chart for each model created. We noticed that the region name, distance to the property, type of property, and number of bedrooms were chosen as the top four important features. Unlike GCP and AWS, Azure has consolidated the ML-related features into Azure ML Services and the data-engineering-related features into Synapse. For basic data integration and transformation requirements, we can also use Azure Data Factory, which helps copy the data from source to target. Azure supports more than 80 source and target connectors.

*7.5. Low-Code ML Platform Based on Databricks*

It is cloud-agnostic, as we have the freedom to choose the data residency of our choice. Data can be stored and hosted on any of the cloud-service-provider ecosystems (AWS, GCP, or Azure). It is a prerequisite to mount the storage on any of the cloud platforms and create a Databricks cluster for computing before trying to ingest the data. When the prerequisites are met, we can easily ingest the data by either providing the dataset's path from the filestore or dragging and dropping the file from the local system. Once the dataset is uploaded to Databricks, we can perform different actions with it. For example, we can create an AutoML job with the given dataset to create an ML model, or we can create a table from the dataset and explore the data by executing a Spark or SQL query against the respective table. In our case, we chose an AWS S3 bucket as a data storage area for our Databricks AutoML experiments.

Configuring the AutoML experiment is smooth with Databricks; we must provision a Databricks cluster for computing, choose the type of problem, choose the dataset, provide the target class (in our case, its price), and name the experiment to keep track of it. Databricks takes care of imputing the missing data if we leave the default Auto option. Apart from the basic details, we can choose the evaluation metric, whether it should be MSE, RMSE, MAE, or R-Squared. In our case, we chose R-Squared as an evaluation metric. Then, we can choose between three different training frameworks recommended by Databricks: LightGBM, Scikit-learn, and XGBoost. We chose all three frameworks for better comparison. We can set the timeout period for how long the experiment should run. Then, it is a good idea to provide a time feature value from the dataset, which should be of the date/time data type; in our case, we chose the date feature. Databricks uses the time feature to split the data into training, testing, and validation sets. We can also provide the data storage location for storing the experiment results in the persistent storage area.

When the AutoML job ends, we have the option of viewing the Python code generated by the Databricks AutoML job for each model in depth by either opening the notebook for the respective model or viewing the data exploration notebook to understand the different data exploratory actions performed by the AutoML job. Models are sorted based on the test_r2 score in descending order, starting from the best model to the model that performed less well.

If we want to obtain more details about a particular auto-generated ML model, then we can simply click on the hyperlink; it will take us to a separate page consisting of end-to-end details about the model description, parameters provided, evaluation metrics considered, and all the artifacts generated during the model creation. It is also possible to register the model from this page, so it can be exposed to the outside world as a Rest API endpoint. We have noticed there are 124 hyperparameters set by the AutoML job; these parameters are constantly tuned by testing different values by generating different models and validating the results against evaluation metrics. In our case, the AutoML job generated more than 100 models within 60 min. Moreover, for each model evaluation metric, artifacts required for deployment and inference were accessible both from the Databricks user interface and in the AWS S3 bucket.

We noticed that all the Databricks-related artifacts are available in the persistent Amazon S3 bucket storage. To deploy the model to an endpoint or containerize it, we can

use the artifacts generated as an AutoML job. We can use the model file in conjunction with the dependent pickle file to deploy the model to any endpoint, the conda file to install the necessary libraries, and the Python environment requirement files to install the necessary Python libraries. Detailed model inference is found by opening the model notebook. This notebook briefly describes importing the required libraries, ingesting data, pre-processing, splitting the data into training/test and validation sets, training the model, generating feature importance, and evaluating the model against different performance metrics.

*7.6. Comparing Models Based on Performance Metrics*

All the auto-generated ML models across GCP, AWS, Azure, and Databricks have been evaluated automatically using different evaluation metrics such as MAE, RMSE, and $R^2$. However, for this comparison, we have only considered the $R^2$ metric, as it indicates how the variance in the independent variable explains the difference in the dependent variable. It is also commonly referred to as the coefficient of determination, which simply determines the variance between the predicted value and the plotted regression line. As we intend to predict the price of the property, we are interested in knowing how the variation in the response variables affects the price of the property. Hence, we chose $R^2$ as a performance metric for comparison.

For the Melbourne housing dataset, the model performance score for the best ML models generated from GCP, AWS, Azure, and the Databricks platform is depicted in Table 4. As per the results, we can see that Azure has the highest $R^2$ score. However, we cannot conclude that one ML platform is better than others based on this metric alone since several other metrics need to be considered, such as the visibility of AutoML jobs operation, traceability of AutoML job logs, the complexity involved in providing inputs for AutoML job, AutoML job elapsed time, customizability, support for co-authoring, how well AutoML can be explained, and how easy it is to deploy a generated ML model. These criteria are briefly explained in the following section.

**Table 4.** Automatic ML Model Performance Score Comparison.

| Model Performance Score | GCP | AWS | Azure | Databricks |
|:---:|:---:|:---:|:---:|:---:|
| $R^2$ | 0.831 | 0.836 | 0.898 | 0.822 |

*7.7. Comparison between ML Platforms*

We created and trained automatic ML models on all three cloud platforms using the respective AutoML services. We noticed similarities and differences in how AutoML services work on each cloud; see Table 5 below.

**Table 5.** Similarities between Cloud ML Platforms.

| Aspect | Similarities |
|:---|:---|
| Prerequisites | Dataset must be created, and data must be uploaded to Cloud Storage for model consumption |
| Feature Importance Results | All three clouds have found similar feature importance results with the best model generated |
| Evaluation Metrics | All three clouds have quite similar evaluation metrics as follows: MAE, MSE, RMSE, and R Squared |

In Table 4, we highlighted the similarities between generating an AutoML model and the three cloud ML platforms. It is a prerequisite for all three ML platforms to upload the data to cloud storage and create a dataset. The best model generated through AutoML jobs from different cloud ML platforms resulted in very similar feature-importance results. Added to that, model evaluation metrics are similar between the three ML platforms. There could also be some additional metrics, but the three cloud ML platforms have considered

commonly used evaluation metrics such as MAE, MSE, RMSE, and $R^2$. On the other hand, the differences are reported in the table below.

In Table 6, we have highlighted some of the differences that we found while generating the AutoML model from the three cloud-based ML platforms. Regarding the traceability aspect, AWS has created all the output logs quite neatly under a directory name, as we provided inside the Amazon S3 bucket. It was quite easy for us to consume the performance logs, Python code for different models, and hyper-parameters tuned from the directory. However, from our experience with GCP and Azure, we found output logs were available in their respective portals; we must manually download them to our local machine if required. Due to the complexity of creating the AutoML model, we found that both AWS and Azure have very few touchpoints for creating the AutoML model and are less time-consuming. However, with GCP, we must provide additional inputs, and it is time-consuming.

**Table 6.** Differences between Cloud ML Platforms.

| AI and ML Features | AWS AutoPilot | GCP AutoML | Azure AutoML |
|---|---|---|---|
| Traceability | All the AutoML operation logs are stored under a given directory in S3 Bucket. | Model-related logs are available only in the UI; however, it is possible to download them. | Model-related logs are available only in the UI; however, it is possible to download them. |
| Complexity | AutoML models can be created at ease with very less touchpoints. | We are expected to provide a few additional inputs for generating AutoML models compared to AWS. | AutoML models can be created at ease with very less touchpoints. |
| Adaptability | Highly adaptable | Not so adaptable. | Somewhat adaptable. |
| Co-Authoring | Yes, with shared compute for all the developers involved. | Yes, with shared compute for all the developers involved. | Yes, but dedicated compute is required for each developer. |
| Explainability | Models created are automatically explained. | Detailed Model summary and explanation are available as a part of UI. | Must provide compute instance or compute cluster manually for explaining each model. |
| Deployment | It can be deployed to internal endpoints. | Possible to containerize the model and deploy it to any endpoint. | Possible to deploy the model to any endpoint. |

Regarding customizing the AutoML job, we found that AWS offered the highest level of customization in comparison to both GCP and Azure. When it comes to the co-authoring feature, where more than one developer can be involved in the ML model development, all three cloud vendors offer this feature. However, only AWS and GCP offer co-authoring with shared compute resources; with Azure, each developer requires dedicated compute resources. When it comes to the explainability of the model, AWS has automatically explained the model without any intervention from our end. With GCP, we can explain the model with a button click. However, with Azure, it is more than a button click, as we have allocated a separate compute instance or cluster for explaining the model. When it comes to deployment, models can be deployed to both internal and external endpoints with GCP and Azure, yet models can only be deployed to the internal endpoint with Amazon.

## 8. Discussion

This research has helped to understand the different data and ML services available across three cloud ecosystems. We demonstrated how to create an ML model automatically without writing a single line of code using Amazon Sagemaker integrated with autopilot, Google Vertex AI integrated with AutoML, and Azure Synapse integrated with ML. During the data-gathering phase, we learned about the different cloud services involved in ingesting the data, transforming the data, training the model, deploying the model, and monitoring the platform. We highlighted the similarities and differences in using AutoML features between the three cloud ML platforms. It was found that AWS Sagemaker offers

better visibility and ease of use when it comes to generating automatic ML-model-related tasks compared to the other two cloud ecosystems. We also found that Databricks offers a simplistic approach for generating automatic ML models and keeping all the heavy lifting under the hood. Google Vertex AI offers AutoML features for a wide number of use cases; this could be an option if we are trying to address a particular problem. The whole practical and theoretical exercise that we conducted as part of this research has exposed us to different data services across major cloud data platforms and on cloud-agnostic data platforms such as Databricks. We recommend that organizations seeking the digitalization of their businesses take advantage of these no-code/low-code ML platforms to speed up analytics implementation and combat the scarcity of human resource challenges. Each cloud platform introduces new data services and features periodically. Hence, it becomes challenging to get a hold of all the new features introduced by the cloud platform immediately. In this case, no-code/low-code ML services come in handy when exploring the possibilities. Another major advantage we noticed is hyperparameter tuning. ML developers or data scientists must have many years of hands-on experience to know the rationale behind every single hyperparameter and how to tune them accordingly. Howbeit, when it comes to automatic ML services, this is taken care of automatically. When it comes to any data science project, a significant amount of time is allocated to ingesting the data and performing data pre-processing tasks. This time is tremendously saved with the help of automatic ML services.

We also think that our lab experiment results were in line with what we obtained from industry experts during the qualitative interviews. We demonstrated how ML platforms are used for data ingestion, ML model building, and evaluation. We also showed how the attained model is self-tuned when it comes to the hyperparameters' configuration. Additionally, we discussed how the obtained models have explainability features. We also discussed the different analytics models available via the platforms and explained their flexibility in selecting and designating features and their roles in the analytics project. We added, discussed, and explained the use of metrics to evaluate models. The use of complicated dataset scenarios and deep learning models are pointers that the interviewees we interviewed have highlighted; however, we have not demonstrated them in this research, which we identify as future research.

### 8.1. Theoretical Implications

Increasingly, data analytics capabilities are seen as crucial to an organization's long-term competitiveness, innovation, and survival [1]. This research study has opened doors for organizations to investigate how the automation of ML analytics projects via low-code platforms could help advance their digitalization strategies as well as their corresponding resource planning. Our study contributes by helping to reduce the barriers between organizations and a quick ML project. The body of literature lacks similar studies articulating and demonstrating how low-code platforms could be used, and, therefore, we see our research as a seed towards further research to investigate the use of low-code platforms further from an integrated perspective, taking into account technical, societal, and financial aspects in a longitudinal fashion.

### 8.2. Practical Implications

This research demonstrated how to conduct analytics projects with the help of low-code ML platforms, which will address human resource scarcity problems and have practical implications in terms of more projects realizing their goals, even in the absence of data scientists within premises. We demonstrated a project that could easily be replicated and help practitioners with insufficient data science skills learn how to conduct the analytics lifecycle with the aid of low-code ML platforms. We demonstrated the foundation of data-driven projects, emphasizing the importance of data and overcoming the resource scarcity problem in a time-efficient manner. We think practitioners need such studies, and

there is a lack of similar ones. Further efforts are still required from the vendor side to provide explainability and reduce operating costs.

## 9. Conclusions

As several businesses have already turned digital and the rest have already started the digital transformation, there is a strong need for some of the digitalization tasks to be automated. In this research, we investigated one of the aspects of digitalization, which is building ML models through ML platform service offerings from three major cloud players: Google, Microsoft, and Amazon. We found that all the cloud vendors have developed advanced ML services through which ML models can be built with ease, without writing a single line of code. At the same time, there is still room for improvement, and we are sure that the cloud vendors will address the most critical aspects of ML platforms, extend ML platform usage to other domain experts apart from ML developers, help organizations make smarter decisions, and help to fast-track digitalization. We do not think automatic ML services will replace data scientists or ML developers. They will, however, complement their ability to perform their tasks effectively and validate their ML model results. Automatic ML services can become our allies if we know when to use them and for what purpose. From this research, we found that automatic ML services, as part of a low-code ML platform, can help to fast-track digitalization, thereby allowing organizations to realize their digitalization goals faster and stay competitive. In this research, we built an automated ML model using a simple tabular dataset for our experiment, but there are also options available to use unstructured and semi-structured data. Due to time limitations and our focus in this research on comparing how an AutoML service is approached by cloud-based and cloud-agnostic ML platforms, validating the other capabilities of the respective ML platforms was not investigated. On the other hand, further research is required pertaining to data security and privacy. There is an opportunity to conduct research by building custom models from the notebook using supported programming frameworks for each ML platform. Furthermore, future research is required regarding more complex scenarios and more versatile datasets. Because most of the beneficiary companies that have implemented an ML platform in either of the cloud ecosystems are not fully utilizing all the capabilities for which they are paying, awareness about the different features of the ML platform needs to be investigated further.

**Author Contributions:** Conceptualization, K.R.R. and A.E.; methodology, K.R.R. and A.E.; experiment, K.R.R. and A.E.; validation, K.R.R. and A.E.; data curation, K.R.R.; writing—original draft preparation, K.R.R.; writing—review and editing, K.R.R. and A.E.; visualization, K.R.R.; supervision, A.E. All authors have read and agreed to the published version of the manuscript.

**Funding:** This research received no external funding.

**Institutional Review Board Statement:** Not applicable.

**Informed Consent Statement:** Not applicable.

**Data Availability Statement:** Not applicable.

**Conflicts of Interest:** The authors declare no conflict of interest.

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
