# Peer review of "Low-Code Machine Learning Platforms: A Fastlane to Digitalization"

_informatics, doi:10.3390/informatics10020050_

Round 1
Reviewer 1 Report
The article presents an exercise consisting to model the Kaggel’s Melbourne housing dataset using 4 platforms AWS, GCP, Azure and Databricks. The idea is to illustrate the capacities of cloud platforms and automatized machine learning solutions to help building, validating and publishing statistical models in an accessible way. The manuscript presents its concepts, methodology and results in a rather clear and pleasant way. However, there are yet several modifications requested before publication, listed below.
Page 3 Table 1 and lines 104-129: Avoid citations in a table. On the other hand, the description of the parameters of the ML methods should be citing either a relevant article or handbook.
Page 3-4 lines 130-154: The authors are repeating themselves. This paragraph can be simplified.
Page 4 line 162: “… to perform several ML tasks these platforms without having programming competencies”. I did not understand the statement. Please rephrase.
Page 5 lines 184-191: The authors introduce an original and valuable contribution through the interviews of several data scientists with various roles. The authors are requested to add a paragraph to develop the methodology of the interview. Do the interviewed persons received the questions in advance? What was the context of the interviews: distance, written, in presence, dedicated room, etc? Did the interviewed persons answered the same questions? Could they proof-read their answers? Etc.
Page 5-6 lines 195-237: The results of the interviews are summarized in a very long bullet point listing which is hardly readable. The authors are requested to at least, organize these into main concepts. Some of these concepts, at first sight, are hardware, software, financial, conceptual. The authors are also invited to describe and properly cite for the concept of MLOps.
Section 5, figure 1-4: The captions of these figures are too short. The authors are invited to develop them a bit to help the reader understand these figures without referring to the text.
Page 6 lines 266-267: “Any (…) change in the [training] set trigger auto-training the [ML model]. If the auto-trained model get low performance score, it is time to build a new or ensemble model.” Here I am confused. It seems to make more sense if the actual model is used retrospectively on the new data and if the performances are not satisfactory, then an autoML is triggered and a new model is trained. The authors are request to clarify.
Page 8 lines 301-301: “… which consists of pipelines and operators of the same.” I did not understood. Please rephrase.
Page 9 line 325: Section 6 title should mention Databricks to be consistent with the other paragraphs in section 5.
Page 9 line 338: “SQL or Spark” I think that it is rather Spark SQL. Please correct if needed.
Page 10 lines 368-379: I think that this paragraph is actually a part of the state of the art and should be placed at the end of the introduction.
Section 7.2, 7.3, 7.4: The first paragraph of each of these sections look like repetition of the section 5. It would improve readability to avoid repetition and complete the section 5 with the information of these short paragraphs.
Page 14 lines 579-581: the performances mentioned here are dedicated to classification but the challenge proposed by the authors is a regression task. So I am confused here.
Page 15 section 7.6: I don’t understand the context of the evaluation of performances of the models. What is the test set? How was it kept isolated while training the models? Is it the same test set for all models? Can the authors provide with a standard deviation of the performance measure?
Page 16 table 5 and 6: the authors provide with interesting tables resuming their experience over the different platforms they have tested. But from the previous sections, some other parts can be compared: the services for preprocessing the data, the setup of the evaluation performances, how models are compared to select the optimal one, etc. It would be nice if the authors can enrich a bit more this part.
Finally, there is factor that is not much mentioned. The AutoML techniques are designed to get the best model according to some optimization criteria. However, the choice an actual ML algorithm is often motivated by pragmatic constrains: small or big data, real time, embarked models, stability of the model, uncertainty, interpretability and many more. How do these platforms answer to these questions or help the end-user to envision his own data mining problem?
In order to answer the above remarks, the major revision are requested for the manuscript.
The English language is fine in general. Some sentences need to be proof-read - I mentioned in the comments to authors, some places that need improvement. But there might be more needs for editing.
Author Response
Dear Reviewer,
thank you very much for taking the time to read out paper, we do appreciate that. The feedback which we have received did help us to make our paper better. We hereby attach a file which includes details of how we have addressed the comments sent to us.
Kind regards,
The authors

Reviewer 2 Report
The popularity of research on automated machine learning platforms is increasing daily. They help a rapid transformation, especially for institutions in the digitalization process. Therefore, the subject of this research is a current and needed issue.
The introduction topic is well-structured and understandable. It should be explained in more detail why the selected cloud software was chosen. It will be more meaningful if it is explained with the support of the literature. How were the questions prepared for the field experts who were interviewed? The answers given by the experts can be thematized based on questions and presented in groups.
The findings section is in a sufficient and understandable structure. The limitations of the study can be added to the conclusion part.
Author Response

(The authors gave the same response as above.)

Round 2
Reviewer 1 Report
The authors did follow most of the recommendations, but not all. However these do not impact critical reading of this contribution. There is no reason to delay this publication.
English quality is acceptable.